# Ultra-Processed Food Consumption Is Associated with an Increased Risk of Abdominal Obesity in Adults: A Cross-Sectional Study in Shanghai

**DOI:** 10.3390/foods14223955

**Published:** 2025-11-18

**Authors:** Wei Lu, Tongxing Ou, Qi Song, Zehuan Shi, Zhuo Sun, Liping Shen, Wenqing Ma, Shupeng Mai, Zhengyuan Wang, Jiajie Zang

**Affiliations:** 1Department of Nutrition and Health, Division of Health Risk Factors Monitoring and Control, Shanghai Municipal Center for Disease Control and Prevention, Shanghai 201107, China; luwei2@scdc.sh.cn (W.L.); songqi@scdc.sh.cn (Q.S.); shizehuan@scdc.sh.cn (Z.S.); sunzhuo@scdc.sh.cn (Z.S.); shenliping@scdc.sh.cn (L.S.); mawenqing@scdc.sh.cn (W.M.); maishupeng@scdc.sh.cn (S.M.); 2School of Public Health, Shanghai University of Traditional Chinese Medicine, Shanghai 201203, China; outongxing2003@163.com; 3School of Medicine, Tongji University, Shanghai 200331, China

**Keywords:** ultra-processed foods, abdominal obesity, adults, cross-sectional study

## Abstract

Ultra-processed foods (UPFs) have become increasingly prevalent in modern diets due to their convenience and variety. These foods typically have higher energy density and lower nutrient density, contributing to adverse health outcomes such as obesity and hypertension. This study aimed to explore UPFs consumption status and its association with health outcomes in a representative Shanghai population, using data from the Shanghai Diet and Health Surveillance. Multi-stage stratified random sampling was adopted, with dietary intake assessed via 3-day consecutive 24 h dietary recalls, and health outcomes evaluated through physical measurements and biochemical indicators. UPFs account for 11.2% of total energy intake overall and 29.0% among high consumers. The overall prevalence of abdominal obesity was 58.9%. After adjusting for confounders, high UPFs consumption was associated with a 28.5% higher risk of abdominal obesity versus non/low consumption (OR = 1.285, 95% CI: 1.059–1.559, *p* = 0.011), with no significant associations with hypertension or diabetes. High UPF consumption is an independent risk factor for abdominal obesity in Shanghai residents, highlighting its public health relevance.

## 1. Introduction

Ultra-processed foods (UPFs) have become an essential part of modern diets due to the increasing demand for convenience and variety in fast-paced lifestyles. UPFs are defined by the NOVA classification system, which categorizes foods into four groups based on the nature, extent, and purpose of industrial processing: Group 1 (Unprocessed or minimally processed foods, e.g., fresh fruits, unprocessed meats); Group 2 (Processed culinary ingredients, e.g., vegetable oils, table sugars, edible salts); and Group 3 (Processed foods), which undergo simple processing techniques such as pickling, fermentation, and thermal treatment. In line with the NOVA system and tailored to Chinese dietary characteristics, the UPFs identified in this study mainly include sugar-sweetened beverages (SSBs), packaged snacks, confectionery, ice cream, chocolate, mass-produced packaged breads, cakes, desserts, biscuits, pastries, pre-prepared pies, pizza, hot dogs, sausages, and other reconstituted meat products. For food items with ambiguous classification, a product was defined as a UPF if its ingredient list contained one or more substances not typically used in home kitchens (e.g., hydrolyzed proteins, mechanically separated meat, fructose, inverted sugar, maltodextrin, interesterified oils, or hydrogenated oils) [1].

UPFs typically have higher energy density and lower nutrient density compared to unprocessed or minimally processed foods [2], and often contain excessive amounts of refined sugars, salt, and unhealthy fats, such as trans and saturated fats [3]. Additionally, they frequently contain high levels of sodium and phosphate-containing flavor enhancers and preservatives [4,5]. High-income countries often exhibit higher consumption levels of UPFs. Previous studies have shown that UPFs can constitute 42% to 60% of total dietary energy intake [6,7,8]. Due to their high energy density and low nutrient content, excessive consumption of UPFs has been frequently associated with weight gain and other adverse health outcomes. The European Prospective Investigation into Cancer and Nutrition included 348,748 adults from 9 European countries. After 5 years of follow-up, higher UPF consumption was positively associated with weight gain. Furthermore, participants in the highest quintile of UPF consumption had a 15% greater risk of becoming overweight or obese compared with those in the lowest quintile during the follow-up period [9]. The French prospective population-based NutriNet-Sante cohort (2009–2019) included 110,260 adult participants. After at least 5 years of follow-up, consumption of UPFs was positively associated with an increase in body mass index (BMI). For every 10% increase in UPF consumption, non-overweight participants had an 11% increased risk of becoming overweight, while non-obese participants had a 9% increased risk of becoming obese [10]. A prospective cohort study involving 22,659 adults with a median follow-up period of 5 years, utilizing data from the UK Biobank, demonstrated that individuals in the highest quartile of UPF consumption had a significantly higher risk of developing both overall and abdominal obesity compared to those in the lowest quartile. Specifically, they had a higher risk of experiencing a ≥5% increase in BMI, waist circumference, and body fat percentage [11]. In addition, the consumption of UPFs has been associated with several health outcomes, including hypertension [12], dyslipidemia [13], metabolic syndrome [14], cardiovascular disease, coronary heart disease, and cerebrovascular disease [15].

The global prevalence of abdominal obesity, a central pathogenic feature of metabolic syndrome, represents a major public health crisis. A key driver of this epidemic may be the escalating consumption of UPFs [16]. Beyond mere caloric excess, emerging evidence indicates that UPFs instigate profound biological alterations that preferentially promote visceral adiposity. These mechanisms include the disruption of appetitive hormones and gut–brain axis signaling, leading to hedonic overeating and impaired satiety [17]. Furthermore, the poor nutritional quality of UPFs, characterized by rapid digestion of refined carbohydrates, induces postprandial hyperglycemia and hyperinsulinemia, creating a metabolic milieu that favors lipid storage and insulin resistance [18].

The majority of these studies have been conducted in high-income countries, such as the United States [12], France [15], Australia [7,19], and Canada [20,21]. However, research on UPFs in China remains limited. Although the consumption of UPFs in China may not be as high as that in high-income countries, it is increasing rapidly [22]. From 2004 to 2023, the number of large-scale food manufacturing enterprises in China increased from 4950 to 10,075, representing a growth rate of 103.5% [23,24]. As an international metropolis, residents in Shanghai can easily access a wide variety of UPFs. Dietary patterns in Shanghai may be more aligned with those of developed nations, potentially resulting in higher UPFs consumption [25]. Therefore, based on data from our Shanghai Diet and Health Surveillance (SDHS) project, which tracks residents’ dietary consumption and health status, we examined the relationship between UPF consumption and health outcomes in a representative Shanghai population.

## 2. Materials and Methods

### 2.1. Study Population and Design

The data for this study were obtained from the SDHS, a representative cross-sectional survey conducted by the Shanghai Municipal Center for Disease Control and Prevention. Launched in 2012, this program has implemented four survey waves to date, with the initial rounds conducted between 2012 and 2014, and subsequent surveys planned at regular intervals to monitor dynamic changes in residents’ dietary patterns and health status. Detailed information on the SDHS’s study design has been described in our previous publication [26,27].

The following formula was employed to estimate the sample size in this survey:$$N=deff\frac{\mu^{2}p(1-p)}{\delta^{2}}$$

The parameters were set as follows: a 95% confidence level (μ = 1.96), an expected obesity prevalence (p) of 22% based on prior metabolic syndrome Shanghai data, a design effect (*deff*) of 2.0 to account for the complex sampling design, and a relative error of 10%. This relative error translates to an absolute margin of error (*δ*) of 0.022 (0.1 × 0.22). The calculation resulted in a required sample size of approximately 2724 participants. To ensure the representativeness of the sample, a multi-stage stratified random sampling method was adopted. Firstly, 48 towns/sub-districts were selected from the city using a population-proportional probability sampling method. Secondly, one to three villages and townships were selected from towns/sub-districts by a simple random sampling method according to the proportion of the population. A total of 72 villages and townships were drawn. Finally, 40 residents were randomly selected from each village/township, stratified by four age groups (18–44 years, 45–59 years, 60–74 years, and ≥75 years), with each group comprising five males and five females. A total of 2880 participants were enrolled following the sampling process.

### 2.2. Data Collection

A standardized questionnaire was employed to collect data on basic personal information, personal health and behavioral habits, as well as dietary intake. The survey was conducted in a one-to-one format by trained investigators. All data were reviewed by the local regional Centers for Disease Control and Prevention (CDC) project team, and then at least 5% of the data was reviewed by the Shanghai CDC project team. If quality control results are “unqualified” by the Shanghai municipal CDC project team, all data should be rechecked by the local district CDC project team.

A 24 h dietary recall was conducted for three consecutive days (two weekdays and one weekend day), during which each participant reported all foods consumed and the place where the foods were consumed over the previous 24 h. Investigators visited each subject’s home and recorded each participant’s diet using a food picture aid and a questionnaire. The questionnaire collected detailed information on the name of the raw material, the amount consumed, and whether the food was ultra-processed.

The detailed data of condiments (including edible oil, table salt, monosodium glutamate, soy sauce, etc.) were weighed by trained investigators using the same brand and model scale. All purchased and wasted condiments were weighed, and consumption was determined based on the changes in condiment stock weight from the beginning to the end of each day. Home condiment weighing was performed the day before and after dinner each night for 4 nights per participant. In addition, the number of people (both family members and guests) who consumed household condiments at each meal was recorded. Personal condiment consumption was calculated as the proportion of total household condiment consumption divided by the proportion of energy consumption of individuals in the household. The sodium intake of condiments for eating out was estimated and converted based on the ratio of energy to the sodium intake of condiments for eating out. Total sodium intake was the sum of sodium intake from cooking salt, monosodium glutamate, soy sauce, ultra-processed foods, and raw foods. The amount of sodium in food was calculated according to the Chinese Food Composition Table, the most authoritative and complete food composition tool in China [28].

Physical measurements were conducted by uniformly trained health workers using standardized methods. The physical examination included assessments of height, weight, blood pressure, waist circumference, and hip circumference. All equipment used for the physical examination had been calibrated and validated for use in the study. Blood pressure was measured using an electronic sphygmomanometer, accurate to 1 mmHg, with three measurements taken 30 s apart.

Collection of blood samples and detection of indicators: A total of 6 mL of fasting venous blood was collected from participants after 10 to 14 h of fasting. The samples were immediately stored in a refrigerator and tested within 8 h. Biochemical parameters measured included fasting blood glucose (FBG) (hexokinase method), high-density lipoprotein cholesterol (HDL-C) (direct method-catalase clearance method), and serum triglycerides (TG) (enzymatic method), all measured using a Hitachi 008α instrument.

### 2.3. The Definition

Participants were stratified into tertiles according to the percentage of total energy intake derived from UPFs over the preceding three days, resulting in three groups: non/low consumption group, medium consumption group, and high consumption group. UPFs in this study are formulations made primarily or entirely from food and additive-derived substances. Most of them have undergone sophisticated industrial processing, such as sweet or savory packaged snacks, confectionery, energy bars, freshly made packaged bread, reconstituted meat products, and pre-prepared frozen or shelf-stable dishes.

BMI was calculated as body weight (kg) divided by the square of height (m^2^). Overweight was defined as 24.0 kg/m^2^ ≤ BMI < 28.0 kg/m^2^, and obesity was defined as BMI ≥ 28.0 kg/m^2^, according to Chinese criteria [29]. Abdominal obesity was defined as a waist circumference (WC) ≥ 90 cm or waist-to-hip ratio (WHR) ≥ 0.90 in men and WC ≥ 85 cm or waist-to-hip ratio (WHR) ≥ 0.85 cm in women based on Chinese criteria [29].

Blood pressure was averaged over three measurements. Hypertension was defined as a history of physician-diagnosed hypertension or systolic blood pressure (SBP) ≥ 140 mmHg or diastolic blood pressure (DBP) ≥ 90 mmHg at the current assessment [30]. Diabetes was defined as a history of physician-diagnosed diabetes or a FBG ≥ 7.0 mmol/L at the current assessment [30]. An individual fulfilling three or more of the following criteria was diagnosed to have metabolic syndrome: (1) WC ≥ 90 cm in men, WC ≥ 85 cm in women. (2) FBG ≥ 6.1 mmol/L. (3) SBP ≥ 135 mmHg or DBP ≥ 85 mmHg. (4) HDL-C < 1.04 mmol/L. (5) TG ≥ 1.7 mmol/L [30].

### 2.4. Statistical Analysis

Categorical variables were summarized as counts (*n*) and percentages (%) in Table 1. Non-normally distributed continuous variables are presented as medians with interquartile ranges (IQR; P_25_–P_75_). Categorical variables, including gender (male/female), smoking status (yes/no), alcohol drinking (yes/no), and other variables, were compared across the three consumption groups (non/low, medium, and high) using contingency table analysis. Group comparisons except those in Table 2 were performed using nonparametric tests: (1) Mann–Whitney U tests for two-group comparisons, (2) Kruskal–Wallis H tests for multi-group comparisons (≥3 groups), with post hoc pairwise analyses conducted using Bonferroni-corrected Mann–Whitney U tests to control for multiple comparisons. Multivariate logistic regression models were used to estimate the odds ratios (ORs) and 95% confidence intervals (95% CI) between UPF consumption and health outcomes. One-way Multivariate Analysis of Variance (One-way MANOVA) was utilized to examine the overall differences in energy and main nutrients intake across three distinct consumption groups (non/low, medium, and high) in Table 2. Prior to analysis, the normality and homogeneity of variance-covariance matrices of the dependent variables (energy and main nutrient intake) were verified, and rank transformation was applied to non-normal data to ensure the robustness of the statistical results [31]. Where the multivariate test was significant, follow-up univariate analysis of variances (ANOVAs) were conducted for each dependent variable, and pairwise group comparisons were performed using Tukey’s HSD. All statistical tests were conducted using a two-sided α level of 0.05, and *p* < 0.05 was considered statistically significant. All statistical analyses were performed using IBM SPSS Statistics version 25.0 (IBM Corp., Armonk, NY, USA) and Python 3.4 (Python Software Foundation, Wilmington, DE, USA). Missing patterns were evaluated using descriptive statistics to classify the missing types into completely random missing, missing at random, and missing not at random. For key variables directly related to the study outcomes (e.g., core indicators of 24 h dietary recall, critical anthropometric parameters), samples with missing values for these variables were excluded to ensure the reliability of the primary analysis. For non-key variables with missing data, appropriate handling methods were adopted based on the missing rate and variable characteristics, including imputation (e.g., mean imputation, multiple imputation), interpolation, and other scientifically validated approaches.

## 3. Results

### 3.1. Baseline Characteristics

Data with critical information missing (*n* = 22) or with a standard person’s daily energy intake of less than 500 kcal (*n* = 7) or more than 5000 kcal (*n* = 9) were excluded. Ultimately, a total of 2842 adults were included in the study, comprising 1414 males (49.8%) and 1428 females (50.2%). Significant differences were observed across the three groups in age distribution, educational attainment, annual household income, marital status, and occupational categories (*p* < 0.05), while no significant differences were found in other indicators (Table 1).

### 3.2. Energy and Nutrient Intake

The median daily energy intake of adult residents in Shanghai is 1620.0 kcal, of which 11.2% comes from UPFs. The intakes of daily total energy and key nutrients were compared across the three groups in Table 2 and Appendix A. One-way multivariate analysis of variance revealed significant differences between the groups (Wilks’ Lambda = 0.8998, F = 16.422, *p* < 0.001) in Appendix A. These differences were statistically significant in energy (F = 10.650, *p* = 0.001), protein (F = 7.594, *p* = 0.006), fat (F = 5.540, *p* = 0.018), cholesterol (F = 12.004, *p* = 0.001), calcium (F = 18.141, *p* < 0.001), iron (F = 12.434, *p* = 0.001), carotene (F = 33.608, *p* < 0.001), thiamine (F = 4.087, *p* = 0.043), riboflavin (F = 5.635, *p* = 0.018), vitamin C (F = 8.701, *p* = 0.003), vitamin E (F = 13.174, *p* = 0.001) and Folic acid (F = 5.844, *p* = 0.016).

Further pairwise comparisons revealed that the high consumption group had a higher daily energy intake (*p* = 0.003) and protein intake (*p* = 0.016) than the non/low consumption group. The daily fat intakes were higher in both the medium and high consumption groups compared to the non/low consumption group (*p* = 0.017; 0.049).

### 3.3. Distribution of Metabolic Disorders Among Different Groups

The prevalence of obesity, overweight or obesity, hypertension, diabetes, and metabolic syndrome was 15.3%, 41.7%, 36.7%, 10.6%, and 19.0%, respectively (Table 3). Notably, the prevalence of abdominal obesity was as high as 58.9%. However, no significant differences were observed in the prevalence of these diseases among the three groups.

### 3.4. Logistic Regression Analysis of Influencing Factors of Metabolic Disorders

Logistic regression analyses were performed to assess the association between UPF consumption and various metabolic disorders, including obesity, overweight and obesity, abdominal obesity, hypertension, diabetes, and metabolic syndrome (Table 4). After adjusting for sex, age, education, annual household income, marital status, occupational status, and daily energy intake, individuals in the high-consumption group had a 28.5% higher risk of abdominal obesity compared to those in the non/low-consumption group (OR = 1.285, 95% CI: 1.059–1.559, *p* = 0.011).

## 4. Discussion

In recent years, the increasing consumption of UPFs and their associated poor nutritional status have garnered widespread attention both domestically and internationally. Numerous foreign studies have shown that excessive consumption of UPFs can lead to obesity, diabetes, hypertension, and cardiovascular diseases [14,21,32]. Large-scale epidemiological studies quantifying UPF consumption remain limited in the Chinese population. Most studies adopt the concept of prepackaged foods, defined as foods that are pre-portioned or prepared in packaging materials and containers for sale by weight or measure, excluding condiments. The consumption rate of prepackaged foods in our study was 82.5%, which is significantly higher than the 59.8% reported among adults in 15 provinces in 2015 by the Chinese Center for Disease Control and Prevention [33]. This phenomenon may be attributed to a complex interplay of multifaceted factors. As an international metropolis, Shanghai has been significantly influenced by Western culture, particularly in terms of dietary habits [34,35]. This cultural influence likely contributes to a higher acceptance of UPFs among its residents. Additionally, the highly developed food industry in Shanghai, combined with an abundant market supply and convenient online and offline platforms, has facilitated the accessibility of UPFs and provided a diverse range of choices. These factors collectively align with the demands of Shanghai residents for food convenience and personalization, which are essential under fast-paced working conditions.

In Shanghai, there is a large population that consumes UPFs, with a notable subset of individuals exhibiting particularly high consumption levels. The proportion of energy intake derived from UPFs was 11.2% (Table 2). In contrast, individuals within the high-consumption group exhibited a significantly higher proportion, with 29.0% of their total energy intake originating from UPFs. The high energy density and low nutrient density of these foods may contribute to excessive caloric intake and fat accumulation. Notably, 5.6% of participants obtained more than 42% of their total energy intake from UPFs, a level close to that observed in Western developed countries [7,36]. Dramatically, this subgroup of participants was relatively younger in age, and female blue-collar workers accounted for a higher proportion of this group.

Multivariable analysis revealed a 28.5% (Table 4) higher risk of abdominal obesity in the high consumption group compared to the non/low consumption group, but similar results were not observed in overweight and obesity. This finding is consistent with the previous studies in some aspects, while some differences remain. A meta-analysis including 9 cross-sectional and 3 cohort studies that were conducted among populations from high-income countries such as the USA, UK, Brazil, Spain, and Canada showed that UPFs consumption was associated with an increased risk of obesity, overweight, and abdominal obesity (OR = 1.55, 1.36, 1.41) [36]. Similar results were also observed in studies from Australia. A study of 7411 Australian adults showed that those in the highest quintile of UPF consumption had a 38% higher risk of abdominal obesity than those in the lowest quintile [19]. Similar perspectives have also been observed in studies from Asian countries. A study of 22,688 Korean adults from the Korea National Health and Nutrition Examination Survey 2016–2020 showed participants in the highest quartile of UPF intake had 19%, 19%, and 13% higher risks for metabolic syndrome, abdominal obesity, and hypertension, respectively, than those in the lowest quartile [37]. The results of the Chinese Food Consumption Survey 2017–2020, which was conducted at 126 investigated sites across 23 provinces, including 38,658 adults aged 18 years and above, regarding the association between overweight and obesity and the consumption of UPFs, were similar to those of our study. It showed that there is no statistically significant difference in the incidence of obesity and overweight between the highest quartile group and the lowest quartile group [38]. Higher UPF intake was associated with an increased risk of abdominal obesity but not overweight or obesity, which may be related to the fact that BMI does not represent the distribution of body fat. Thus, its ability to identify risk factors associated with metabolic and cardiovascular disease is weaker than that of abdominal obesity [39]. It is noteworthy that Asians are more likely to exhibit the “metabolically obese” phenotype, characterized by central obesity and abdominal fat accumulation despite having a normal body weight and BMI. Under the same BMI conditions, Asians had a higher body fat content than Westerners [40,41]. Studies have shown that abdominal adipose tissue is more metabolically active and is a key determinant of metabolic abnormalities contributing to the risk of obesity-related diseases [42]. Excessive energy intake is an important risk factor for the increased incidence of abdominal obesity, which is strongly influenced by dietary energy density [43]. UPFs often have a high energy density and a low nutrient density. This study also observed that individuals in the high-consumption group had significantly higher intakes of energy and fat compared to those in the low-consumption group, while their intakes of iron, vitamin C, vitamin E, and niacin were significantly lower. Notably, the intakes of carotene, vitamin C, vitamin E, and niacin among the high UPF consumption group all failed to meet the Recommended Nutrient Intakes (RNIs) specified in the Dietary Reference Intakes for Chinese Residents (2023). This consistent micronutrient deficit, coupled with excessive energy and fat intake, further underscores the health risk associated with high UPF consumption in Shanghai.

While epidemiological studies from Western populations with high dietary reliance on UPFs have consistently demonstrated associations between UPF consumption and metabolic disorders (e.g., hypertension and type 2 diabetes), our analysis in the Shanghai cohort did not replicate these associations. The generally younger age profile of the high UPF consumption group may act as a protective factor against these conditions, thus obscuring the potential long-term adverse effects of high UPF intake. On the other hand, disparities in both the absolute quantity of UPFs consumption and the specific product types may also underpin the divergent results.

This finding is consistent with the result of the study using data from 5147 adults in the China Health and Nutrition Survey cohort. They found that the risk of metabolic syndrome and abdominal obesity increased by 17% and 33% in the highest quartile with UPFs consumption, with the lowest quartile as a reference, but no correlation was observed between UPFs consumption and raised fasting plasma glucose [1]. Although we did not observe associations with hypertension, diabetes, and metabolic syndrome, individuals with higher UPF consumption were more likely to have abdominal obesity. Numerous studies have shown that abdominal obesity is an independent risk factor for hypertension, diabetes, and metabolic syndrome [44,45].

The multi-stage stratified random sampling method was adopted in this study to ensure the representativeness of the samples. Questionnaire collection and physical measurements were conducted by trained investigators on a one-to-one basis with strict quality control to minimize error. Additionally, we weighed various condiments to enhance the accuracy of the dietary intake data. Our study also has some limitations. As a cross-sectional study, our analysis establishes associations but cannot determine causal relationships between UPF consumption and health outcomes. Although some potential confounders were adjusted, recall bias from residual confounders, such as the 24 h dietary recall survey, could not be completely avoided.

## 5. Conclusions

Our study revealed a high prevalence of UPF consumption among Shanghai residents, with 80.2% reporting regular intake. Notably, 5.6% of participants obtained ≥ 42% of their total daily energy from UPFs—a level comparable to Western populations. The study provided preliminary evidence for the association between high UPF consumption and increased abdominal obesity risk. Given that abdominal obesity is a sensitive indicator of fat accumulation and an independent risk factor for hypertension, diabetes, and metabolic syndrome, the consumption of UPFs deserves ongoing attention in Shanghai. We recommend that the Chinese government prioritize the regulation and education regarding UPF consumption to mitigate potential health risks.

## Figures and Tables

**Table 1 foods-14-03955-t001:** Baseline characteristics of the study population according to the consumption of ultra-processed foods.

	Total	Non/Low Consumption Group	Medium Consumption Group	High Consumption Group	*χ* ^2^	*p*-Value
**N**	2842 (100.0) *	949 (33.4)	945 (33.3)	948 (33.4)		
**Sex**						
Male	1414 (49.8)	484 (51.0)	470 (49.7)	460 (48.5)	1.165	0.559
Female	1428 (50.2)	465 (49.0)	475 (50.3)	488 (51.5)		
**Age, years**						
18–44	712 (25.1)	211 (22.2)	222 (23.5)	279 (29.5)	20.453	0.002
45–59	707 (24.9)	230 (24.2)	249 (26.3)	228 (24.1)		
60–74	727 (25.6)	260 (27.4)	256 (27.1)	211 (22.3)		
≥75	695 (24.5)	248 (26.1)	218 (23.1)	229 (24.2)		
**Education status**						
Junior high school and below	1294 (45.5)	525 (55.4)	415 (43.9)	354 (37.3)	72.088	<0.001
High school or secondary vocational school	635 (22.3)	193 (20.4)	223 (23.6)	219 (23.1)		
Junior college and above	912 (32.1)	230 (24.3)	307 (32.5)	375 (39.6)		
**Family income last year, RMB**						
< 100,000	973 (34.2)	364 (38.6)	314 (33.4)	295 (31.4)	15.002	0.005
100,000–200,000	1124 (39.5)	350 (37.2)	398 (42.3)	376 (40.1)		
≥200,000	724 (25.5)	228 (24.2)	229 (24.3)	267 (28.5)		
**Marital status**						
Single	228 (8.0)	56 (5.9)	69 (7.3)	103 (10.9)	17.177	0.002
Married/cohabiting	2201 (77.4)	752 (79.2)	742 (78.5)	707 (74.6)		
Divorce, widowhood, or other	413 (14.5)	141 (14.9)	134 (14.2)	138 (14.6)		
**Occupation status**						
Mental activity mainly	611 (21.5)	218 (23.0)	200 (21.2)	193 (20.4)	56.853	<0.001
Physical activity mainly	497 (17.5)	118 (12.4)	146 (15.4)	233 (24.6)		
Retirement	1311 (46.1)	469 (49.4)	463 (49.0)	379 (40.0)		
Unemployment and other	423 (14.9)	144 (15.2)	136 (14.4)	143 (15.1)		
**Smoking status**						
Yes	778 (27.4)	265 (27.9)	263 (27.8)	250 (26.4)	0.723	0.697
No	2064 (72.6)	684 (72.1)	682 (72.2)	698 (73.6)		
**Drinking status**						
Yes	902 (31.7)	305 (32.1)	296 (31.4)	301 (31.8)	0.134	0.935
No	1938 (68.2)	644 (67.9)	648 (68.6)	646 (68.2)		

*: The numerical values inside the parentheses denote the proportions (%), and the numerical values outside the parentheses denote the number of objects.

**Table 2 foods-14-03955-t002:** Energy and main nutrients intake and sources.

	Total (*n* = 2842)		Non/Low Consumption Group (*n* = 949)		Medium Consumption Group (*n* = 945)		High Consumption Group (*n* = 948)		F ^#^	*p*-Value
	Intake	UPF (%)	Intake	UPF (%)	Intake	UPF (%)	Intake	UPF (%)		
**Energy, kcal/d**	1620.0 (1278.0, 2035.6) *	11.2 (1.7, 22.8)	1568.5 ^a^ (1223.1, 1998.8)	0.0 (0.0, 1.7)	1624.8 ^a,b^ (1286.4, 2062.9)	11.2 (8.0, 14.3)	1652.3 ^b^ (1324.2, 2038.7)	29.0 (22.8, 37.2)	10.650	0.001
**Protein, g/d**	69.3 (53.5, 91.3)	12.3 (5.6, 22.7)	66.8 ^a^ (50.8, 90.6)	0.0 (0.0, 1.5)	70.8 ^a,b^ (54.5, 91.9)	9.1 (6.1, 13.3)	70.6 ^b^ (55.3, 91.0)	24.9 (17.7, 34.3)	7.595	0.006
**Fat, g/d**	61.7 (45.8, 85.2)	10.0 (3.5, 20.6)	60.0 ^a^ (41.9, 85.3)	0.0 (0.0, 0.4)	63.0 ^b^ (47.3, 86.0)	7.2 (4.0, 12.0)	61.6 ^b^ (48.2, 84.6)	22.3 (14.5, 32.9)	5.540	0.019
**Carbohydrate, g/d**	181.5 (139.4, 241.6)	18.6 (7.4, 32.5)	176.1 (132.5, 241.2)	0.0 (0.0, 0.8)	177.8 (137.3, 237.1)	13.6 (8.6, 19.9)	192.6 (147.5, 245.8)	35.4 (27.1, 47.4)	2.283	0.131
**Cholesterol, mg/d**	456.8 (305.3, 631.3)	0.0 (0.0, 5.3)	443.9 ^a^ (285.0, 623.8)	0.0 (0.0, 0.0)	480.0 ^a^ (321.3, 651.3)	0.2 (0.0, 4.3)	454.0 ^b^ (311.9, 614.5)	4.8 (0.1, 16.9)	12.005	0.001
**Dietary fiber, g/d**	6.0 (4.2, 9.3)	9.1 (1.3, 23.6)	6.1 (4.0, 9.7)	0.0 (0.0, 0.8)	6.1 (4.3, 9.1)	6.8 (1.0, 15.6)	6.0 (4.3, 8.9)	21.2 (8.4, 39.1)	0.013	0.911
**Calcium, mg/d**	444.7 (314.2, 606.0)	8.3 (3.2, 20.0)	408.5 ^a^ (287.2, 575.8)	0.0 (0.0, 2.0)	461.0 ^b^ (324.1, 629.9)	6.1 (2.8, 12.2)	468.1 ^b^ (321.9, 621.5)	17.9 (8.5, 32.4)	18.141	<0.001
**Iron, mg/d**	17.8 (13.8, 23.7)	8.8 (3.5, 17.6)	18.3 ^a^ (14.1, 24.5)	0.0 (0.0, 1.4)	18.1 ^a^ (13.8, 23.8)	6.9 (3.8, 10.9)	17.1 ^b^ (13.3, 22.6)	18.5 (11.1, 27.8)	12.434	<0.001
**Phosphorus, mg/d**	874.4 (685.2, 1131.9)	10.8 (4.4, 21.4)	843.8 (663.9, 1130.1)	0.0 (0.0, 1.6)	906.6 (699.8, 1156.4)	8.7 (5.1, 13.3)	873.7 (701.5, 1110.3)	23.1 (14.1, 32.6)	1.571	0.210
**Potassium, mg/d**	1604.6 (1241.1, 2089.9)	9.7 (3.9, 20.3)	1560.1 (1194.2, 2085.9)	0.0 (0.0, 1.9)	1645.5 (1272.8, 2107.1)	7.6 (4.2, 12.9)	1612.4 (1243.6, 2074.9)	21.1 (12.3, 31.4)	1.494	0.222
**Sodium, mg/d**	3730.9 (2683.2, 5261.6)	6.1 (1.8, 14.5)	3725.5 (2628.9, 5365.5)	0.0 (0.0, 0.8)	3732.2 (2783.0, 5354.1)	4.0 (1.4, 9.1)	3732.6 (2643.9, 5107.9)	12.1 (6.1, 21.8)	0.051	0.821
**Vitamin A, mg/d**	422.7 (290.5, 608.5)	2.3 (0.0, 11.1)	426.1 (292.1, 639.0)	0.0 (0.0, 0.3)	431.2 (302.5, 600.7)	1.3 (0.0, 6.4)	407.4 (278.5, 586.4)	7.2 (0.7, 20.4)	1.057	0.304
**Carotene, mg/d**	1532.3 (881.9, 2416.1)	0.1 (0.0, 4.7)	1713.0 ^a^ (1013.5, 2704.3)	0.0 (0.0, 0.0)	1521.8 ^b^ (946.0, 2380.4)	0.0 (0.0, 2.4)	1357.4 ^c^ (770.2, 2222.6)	1.3 (0.0, 11.1)	33.609	<0.001
**Thiamine, mg/d**	0.7 (0.5, 1.0)	8.2 (2.7, 20.7)	0.7 (0.5, 1.0)	0.0 (0.0, 0.9)	0.7 (0.5, 1.0)	6.4 (3.0, 11.8)	0.7 (0.5, 1.0)	20.6 (10.3, 35.3)	4.087	0.053
**Riboflavin, mg/d**	0.8 (0.6, 1.1)	8.6 (3.3, 18.1)	0.8 ^a^ (0.6, 1.0)	0.0 (0.0, 1.4)	0.8 ^b^ (0.6, 1.1)	6.2 (3.3, 11.8)	0.8 ^b^ (0.6, 1.1)	18.0 (10.4, 29.7)	5.636	0.018
**Niacin, mg/d**	13.5 (10.0, 18.4)	8.4 (2.4, 18.5)	13.6 (10.0, 18.6)	0.0 (0.0, 0.6)	13.7 (10.2, 18.3)	6.3 (2.9, 11.4)	13.2 (9.9, 18.1)	19.0 (10.8, 32.3)	2.317	0.128
**Vitamin C, mg/d**	60.3 (40.6, 91.7)	0.0 (0.0, 4.3)	64.0 ^a^ (41.0, 95.6)	0.0 (0.0, 0.0)	60.8 ^a^ (41.9, 94.0)	0.0 (0.0, 1.6)	56.7 ^b^ (38.1, 85.6)	1.0 (0.0, 13.7)	8.701	0.003
**Vitamin E, mg/d**	18.8 (12.0, 28.4)	5.7 (1.5, 13.7)	19.3 ^a^ (12.0, 30.1)	0.0 (0.0, 0.3)	19.6 ^a^ (12.7, 28.9)	4.3 (1.7, 8.6)	17.9 ^b^ (11.1, 26.6)	12.2 (6.0, 25.6)	13.174	0.001
**Folic acid, mg/d**	48.1 (7.7, 110.5)	0.0 (0.0, 5.4)	56.3 ^a^ (7.4, 118.1)	0.0 (0.0, 0.0)	47.9 ^a,b^ (7.3, 110.5)	0.0 (0.0, 2.4)	41.6 ^b^ (9.0, 94.8)	0.0 (0.0, 23.4)	5.844	0.016
**Overall**									16.423	<0.001

*:The numerical values inside the parentheses denote the 25th and 75th percentiles (P_25_, P_75_), and the numerical values outside the parentheses denote the median; ^a, b, c^: The same letter appearing for two groups denotes the absence of significant between-group differences, whereas different letters for different groups denote the presence of significant between-group differences (*p* < 0.05); ^#^: Rank transformation was applied to non-normal data to ensure the robustness of the statistical results. One-way Multivariate Analysis of Variance (One-way MANOVA) was utilized to examine the overall differences in energy and main nutrients intake across three distinct consumption groups (non/low, medium, and high). Where the multivariate test was significant, follow-up univariate ANOVAs were conducted for each dependent variable, and pairwise group comparisons were performed using Tukey’s HSD.

**Table 3 foods-14-03955-t003:** The distribution of Metabolic Disorders in people with different consumption of ultra-processed foods.

	Total (*n* = 2842)	Non/Low Consumption (*n* = 949)	Medium Consumption (*n* = 945)	High Consumption (*n* = 948)	*χ* ^2^	*p*-Value
**Obesity**	434 (15.3) *	151 (16.1)	147 (15.7)	136 (14.5)	1.033	0.597
**Overweight and obesity**	1185 (41.7)	410 (43.5)	394 (42.0)	381 (40.3)	1.993	0.369
**Abdominal obesity**	1673 (58.9)	549 (57.9)	543 (57.5)	581 (61.3)	3.470	0.176
**Hypertension**	1036 (36.7)	364 (38.6)	341 (36.3)	331 (35.0)	2.726	0.256
**Diabetes**	300 (10.6)	100 (10.5)	88 (9.3)	112 (11.8)	3.139	0.208
**Metabolic syndrome**	539 (19.0)	186 (19.8)	179 (19.1)	174 (18.4)	0.543	0.762

*: The numerical values inside the parentheses denote the proportions (%), and the numerical values outside the parentheses denote the number of objects.

**Table 4 foods-14-03955-t004:** Logistic regression analysis of influencing factors of various diseases.

	Non/Low Consumption Group	Medium Consumption Group			High Consumption Group		
		OR	*p*	95% CI	OR	*p*	95% CI
**Obesity**							
Crude	Reference	0.974	0.832	0.760–1.247	0.883	0.331	0.686–1.135
Model 1	Reference	0.971	0.816	0.757–1.245	0.860	0.244	0.667–1.108
Model 2	Reference	0.983	0.894	0.764–1.264	0.897	0.414	0.692–1.164
**Overweight and obesity**							
Crude	Reference	0.938	0.493	0.781–1.126	0.877	0.158	0.730–1.053
Model 1	Reference	0.946	0.557	0.787–1.138	0.893	0.232	0.742–1.075
Model 2	Reference	0.977	0.809	0.811–1.178	0.938	0.513	0.776–1.135
**Abdominal obesity**							
Crude	Reference	0.984	0.864	0.820–1.181	1.153	0.127	0.960–1.386
Model 1	Reference	1.007	0.938	0.837–1.213	1.246	0.022	1.033–1.504
Model 2	Reference	1.027	0.784	0.851–1.239	1.285	0.011	1.059–1.559
**Hypertension**							
Crude	Reference	0.905	0.297	0.751–1.091	0.856	0.104	0.710–1.032
Model 1	Reference	0.946	0.584	0.777–1.153	0.946	0.583	0.775–1.154
Model 2	Reference	0.983	0.867	0.804–1.202	0.991	0.929	0.806–1.218
**Diabetes**							
Crude	Reference	0.872	0.373	0.645–1.179	1.137	0.378	0.854–1.514
Model 1	Reference	0.903	0.511	0.665–1.225	1.232	0.159	0.921–1.647
Model 2	Reference	0.926	0.623	0.680–1.260	1.241	0.159	0.919–1.675
**Metabolic syndrome**							
Crude	Reference	0.960	0.725	0.764–1.206	0.917	0.461	0.729–1.154
Model 1	Reference	0.996	0.971	0.788–1.258	0.990	0.934	0.783–1.253
Model 2	Reference	1.025	0.838	0.809–1.298	1.054	0.670	0.827–1.344

OR = Odds Ratio; 95%CI = 95% Confidence Interval. Model 1: Adjusted for sex and age. Model 2: Adjust Model 1 plus education status, annual household income, marital status, occupation status, and daily energy intake.

## Data Availability

The original contributions presented in the study are included in the article/Appendix A; further inquiries can be directed to the corresponding authors.

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
