# Peer review of "Ultra-Processed Food Consumption Is Associated with an Increased Risk of Abdominal Obesity in Adults: A Cross-Sectional Study in Shanghai"

_foods, 2025, doi:10.3390/foods14223955_

Round 1
Reviewer 1 Report
Comments and Suggestions for Authors
I have reviewed the paper entitled: “Ultra-processed food consumption is associated with an increased risk of abdominal obesity in adults: A cross-sectional study in Shanghai”.
This study was aimed to investigate the consumption status of ultra-processed food (UPFs) and their association with health outcomes in a representative population in Shanghai. Dietary intake was assessed using a 24-hour dietary recall over three consecutive days (two weekdays and one weekend day). Physical measurements and biochemical indicators were collected to evaluate health outcomes.
It is the original research paper, well designed and conducted, the obtained results are interesting, with good contribution to the field. Only minor changes in the structure of some sentences and some typographical and grammatical corrections are needed. My suggestion is to make minor revisions.
There are some suggestions for the study:
Abstract is too long. According to the Instructions for authors, it is limited to 200 words maximum. Authors should shorten the abstract in accordance with the technical requirements of the Journal.
Introduction provides relevant theoretical background, statement of the problem and proposed approach.
Section Materials and methods is sufficiently detailed described.
The results are abundant, well presented in the tables, with good contribution to the field. However, the authors must precisely define how the values ​​in Table 1 are given. Letters indicating differences between groups are missing in Table 3. The authors should define in Table number 4 what the abbreviations OR and CI represents for easier readability. The discussion interprets the findings in the line with obtained results and with those from the literature.
The authors should approach conclusions with caution. This is a cross-sectional study, analysis establishes associations but cannot determine causal relationships between ultra-processed food consumption and health outcomes. Further studies, such as intervention trials and longitudinal studies, are needed to identify the mechanisms of correlation between consumption of UPFs and health-related outcomes in a representative Shanghai adult population.
The style of references should be in accordance with the technical requirements of the Foods.
Comments on the Quality of English LanguageThe quality of English language is relatively good. Redundancy should be avoided, as well as unclear and too long sentences that make manuscript difficult to read and understand. Changes in the structure of some sentences and some typographical and grammatical corrections are needed. In this sense, it could be useful to have the help of colleague who is more experienced in writing scientific papers.
Reviewer 2 Report
Comments and Suggestions for Authors
Please find all comments and edits in the attached PDF file.

Reviewer 3 Report
Comments and Suggestions for Authors
Dear authors,
Your study provides evidence on a topic of broad interest across various fields of knowledge. However, after a review focused on the statistical analysis and results section, I believe you should make some adjustments.
1. You indicate that you performed the chi-square test (Table 1). You should correct and explain that you used contingency tables of two rows and four columns, for example, male and female for groups of low, medium, and high consumption. Adjust and correct this, as there are various chi-square tests (see [reference]).
2. You indicate that you performed non-parametric ANOVA tests (Tables 2 and 3). The argument based on not adhering to a normal error distribution is appropriate. However, performing 19 separate analyses increases the risk of Type I errors and could lead to Type II errors. Therefore, it is appropriate to perform a one-way MANOVA, and in this regard, you are required to indicate the estimated parameters: Wilks' lambda, partial eta-squared, and you can obtain the estimate of the partial correlation matrix (interrelationships between the dependent variables).
3. The analysis requires compliance with the assumptions of normal error distribution and homogeneity of variances, for which you can transform your data into a "rank" distribution using the technique suggested by Conover and Iman (see reference) and apply the procedure.
Consequently, you need to reanalyze your data and indicate the average values ​​with their standard errors from your MANOVA, providing evidence based on a well-applied statistical technique. This will have a reference point based on the reproducibility and repeatability of the method, aspects of great importance in science.
Sincerely,
Comments on the Quality of English LanguageThe English could be improved
Reviewer 4 Report
Comments and Suggestions for Authors
“Ultra-Processed Food Consumption Is Associated with an Increased Risk of Abdominal Obesity in Adults: A Cross-Sectional Study in Shanghai”
This manuscript presents a timely and relevant cross-sectional analysis of ultra-processed food (UPF) consumption and its association with metabolic disorders among a representative adult population in Shanghai, China. The topic is of significant public health importance, especially given the rapid nutritional transition in China. The study is well-designed, employing a robust multi-stage stratified sampling method, detailed dietary assessment (3-day 24-hour recall), and rigorous quality control measures. The finding of a 28.5% increased risk of abdominal obesity in high UPF consumers, independent of key confounders, is a valuable contribution to the literature, particularly from an Asian context where such data are limited. The following modifications are suggested:
- Clarity in UPF Definition and Classification: The manuscript would benefit from a more detailed description of how UPFs were specifically identified and classified within the Chinese dietary context. While the NOVA system is referenced, providing a list of specific food categories (e.g., sugary beverages, instant noodles, packaged snacks, reconstituted meats) considered as UPFs in this study would enhance reproducibility and clarity.
- Interpretation of Null Findings: The discussion appropriately notes the lack of association with hypertension, diabetes, and metabolic syndrome. However, this point could be further strengthened. It would be useful to speculate more explicitly on potential reasons. For instance, could the overall younger age of the high-consumption group (Table 1) be a protective factor against these conditions, masking a potential long-term effect? Or is the "metabolically obese" phenotype (discussed later) a precursor, meaning that with longer exposure or follow-up, these associations might emerge? Elaborating on these possibilities would provide a more significant interpretation.
- Contextualization of UPF Intake: The result that 5.6% of participants have UPF-derived energy levels (>42%) comparable to Western countries is striking. It would be helpful to briefly discuss the potential socio-economic or demographic characteristics of this specific sub-group. Are they younger, more urban, of higher income? A sentence or two linking this back to the baseline characteristics in Table 1 would be insightful.
Reviewer 5 Report
Comments and Suggestions for Authors
The authors evaluated "Ultra-Processed Food Consumption and its Association with an increased Risk of Abdominal Obesity in Adults", It is well written concise study, however I have few suggestion to improve the manuscript.
- Introduction: After introductory paragraph, it would be better if authors add 1 paragraph considering how UPF contribute for biological or hormonal changes specially linked with abdominal obesity in different population groups.
- Write the method for sample size calculation.
- In methodology section, plz explain about any missed data (how authors addressed that issue). Moreover, the authors collected data based on the 24 hours recall however, 24 recall method along with 3 days dietary record or with food frequency data can present better results with minimum biases. this should be considered as a study limitation.
- Results are well presented
- In discussion section compare population nutrient intake such as calcium, sodium, energy etc with the DRI.
Round 2
Reviewer 3 Report
Comments and Suggestions for Authors
Dear authors,
After reviewing the responses to the points raised in the previous review, I must comment that the technique for avoiding violations of the assumptions of normal error distribution and homogeneity of variances in MANOVA or other generalized linear models is to apply the method suggested by Conover, W. J., & Iman, R. L. (1981). Rank transformations as a bridge between parametric and nonparametric statistics. The American Statistician, 35(3), 124-129. In short, it is a combination of parametric and nonparametric methods. In this sense, what is required is to perform MANOVAs with the data on a rank scale and report the estimated parameters indicated in the previous review, and then perform another analysis with the original scale of the data to report the average values ​​and standard errors. It is important to note that the assumptions do not need to be verified for the rank data because this is the procedure used in nonparametric tests.
Sincerely,
Round 3
Reviewer 3 Report
Comments and Suggestions for Authors
Dear Authors,
After reading and reviewing your manuscript, I believe you have made the necessary adjustments. It is important to note that statistical analyses form the core of any research and require reproducibility and repeatability, key elements that many studies have been shown to lack. The changes you made to your study address these requirements.
Sincerely,